# *GmFAD3A,* A *ω*-3 Fatty Acid Desaturase Gene, Enhances Cold Tolerance and Seed Germination Rate under Low Temperature in Rice

**DOI:** 10.3390/ijms20153796

**Published:** 2019-08-03

**Authors:** Xin Wang, Chao Yu, Yi Liu, Lu Yang, Yang Li, Wen Yao, Yicong Cai, Xin Yan, Shaobo Li, Yaohui Cai, Shaoqing Li, Xiaojue Peng

**Affiliations:** 1Key Laboratory of Molecular Biology and Gene Engineering of Jiangxi Province, College of Life Science, Nanchang University, Nanchang 330031, China; 2National Key Laboratory of Wheat and Maize Crop Science, College of Life Sciences, Henan Agricultural University, Zhengzhou 450002, China; 3Key Laboratory of Crop Physiology, Ecology, and Genetic Breeding, Ministry of Education, College of Agronomy, Jiangxi Agricultural University, Nanchang 330045, China; 4Jiangxi Super-Rice Research and Development Center, Nanchang 330200, China; 5State Key Laboratory of Hybrid Rice, College of Life Sciences, Wuhan University, Wuhan 430072, China

**Keywords:** *Oryza sativa*, *ω*-3 fatty acid desaturase, *GmFAD3A*, cold stress, seed germination

## Abstract

Low temperature is an environmental stress factor that is always been applied in research on improving crop growth, productivity, and quality of crops. Polyunsaturated fatty acids (PUFAs) play an important role in cold tolerance, so its genetic manipulation of the PUFA contents in crops has led to the modification of cold sensitivity. In this study, we over-expressed an *ω-3* fatty acid desaturase from *Glycine max* (*GmFAD3A*) drove by a maize ubiquitin promoter in rice. Compared to the wild type (ZH11), ectopic expression of *GmFAD3A* increased the contents of lipids and total PUFAs. Seed germination rates in *GmFAD3A* transgenic rice were enhanced under low temperature (15 °C). Moreover, cold tolerance and survival ratio were significantly improved in *GmFAD3A* transgenic seedlings. Malondialdehyde (MDA) content in *GmFAD3A* transgenic rice was lower than that in WT under cold stress, while proline content obviously increased. Meanwhile, the activities of superoxide dismutase (SOD), hydroperoxidase (CAT), and peroxidase (POD) increased substantially in *GmFAD3A* transgenic rice after 4 h of cold treatment. Taken together, our results suggest that *GmFAD3A* can enhances cold tolerance and the seed germination rate at a low temperature in rice through the accumulation of proline content, the synergistic increase of the antioxidant enzymes activity, which finally ameliorated the oxidative damage.

## 1. Introduction

Rice (*Oryza sativa* L.) is an important food crop that feeds more than half of the world’s population [1]. Global food security is being challenged by the convergence of multiple factors, including the continuously growing population, reduced arable land, demand for biofuel production, and abiotic stress [2]. Using genetic engineering biotechnology to improve cold-stress resistance of rice is an alternative strategy. This includes using genetic engineering biotechnology to improve the cold resistance of rice.

Low temperature is one of several environmental stress factors that are applied in research aimed at improving the growth, productivity, and quality of crops [3]. To adapt to the stress, plants have developed many ways to balance the effects of stress-induced damage, such as increasing the contents of proline, the activity of detoxifying substances or enzymes such as superoxide dismutase (SOD), and peroxidase (POD) [4]. Cell membranes are major sites of freezing injury and cold acclimation. Membrane fluidity is important to sustain the functional activity of membrane proteins and the membranes themselves and is directly affected by temperature [5]. Lipids are an essential component of cell membranes. Plant seeds store lipids as a food reserve for germination and seedling growth. The predominant component of seed lipids, triacylglycerols (TAGs) are not only essential for human nutrition, but also valuable as feedstocks for various industrial products and biofuels [6]. In plants, the level of polyunsaturated fatty acids (PUFAs) are essential for cold acclimation and is essential for the regulation of cholesterol synthesis and transportation for the maintenance of cellular membranes [7,8].

PUFAs are essential for cold stress. Regulating the expression of the PtFAD2 enzymes could potentially alter PUFAs content in membrane lipids [7]. Genetic manipulation of the levels of PUFAs has led to the modification of cold sensitivity in tobacco plants [9]. Changes in freezing tolerance in hybrid poplar was caused by up- or downregulation of *PtFAD2* gene expression [7]. PUFAs are the main component of soybean oil [10]. Genes controlling the oleic acid and PUFA contents in soybean seed oil have been characterized, containing the two oleate desaturase genes GmFAD2-1A and GmFAD2-1B and three linoleate desaturase genes GmFAD3A, GmFAD3B, and GmFAD3C [11,12]. Mutation of *GmFAD3* resulted in lower linolenic acid content (from 7% to 10%) [10,13]. A tobacco FAD3 expressed in rice could increase α-linolenic acid (ALA, C18:3) level up to 2.5-fold [14]. The ALA content was increased up to a 13-fold when soybean FAD3 driven by the maize ubiquitin-1 promoter was introduced in rice [15]. However, the function of *GmFAD3A* to enhance rice to confer cold tolerance is still unclear.

In this study, the *GmFAD3A* was allogenous expressed in rice. The ectopic expression of *GmFAD3A* enhances cold stress tolerance in rice, including seed germination rates at low temperature (15 °C) and cold tolerance at the seeding stage. We also evaluated lipid content, the malondialdehyde (MDA), proline content, superoxide dismutase (SOD), and peroxidase (POD) activities to explain its cold stress tolerance.

## 2. Results

### 2.1. Structural Characteristics and Cladogram of GmFAD3A

Sequence analysis indicated that *GmFAD3A* (Glyma.14G194300) contained eight exons with a 4, 229-bp genomic sequence (Figure 1a). Annotation of the protein sequence on the Pfam website revealed that amino acids 3–67 and 72–333 of the GmFAD3A protein respectively represented domain of unknown function (DUF) and FA_desaturase conserved domains (Figure 1b), implying that GmFAD3A encodes a fatty acid desaturase.

To compare the evolutionary relationship of FAD homologs in rice and soybean, a Neighbor-Joining phylogenetic tree of 14 proteins with high similarity was constructed (Figure 1c). Nine homologs of FADs were identified in rice. Two GmFAD3A homologs of rice (OsFAD8 and OsFAD7) were the closest to GmFAD3A and were categorized into the same clade including GmFAD3A (Figure 1c). When compared with OsFAD8 and OsFAD7, the identity of GmFAD3A amino acid sequences was 72% in both cases.

### 2.2. Ectopic Expression of GmFAD3A in Rice

In order to study the function of *GmFAD3A* in rice, the gene was cloned from soybean and then transformed into *Oryza sativa* L. ssp. *japonica* cv. Zhonghua 11 (ZH11) by constitutive ubiquitin promoter via an *Agrobacterium tumefaciens*-mediated transformation. More than 40 independent transgenic plants were obtained and most (>90%) were positive transgenic plants detected by hygromycin resistance gene (*Hpt*) PCR analysis (Appendix A). Four lines (OE4, OE6, OE8, and OE10) with high expression levels compared to ZH11 were selected for going down to the next generation (Appendix A). Furthermore, two independent transgenic lines (OE4-2 and OE8-5) with high expression levels originating in the T_0_ generations (OE4 and OE8) were advanced up to the T_2_ generation (Figure 2a,b). All the T_2_ progeny transgenic seeds of OE8-5 lines can grow from hygromycin solution. In addition, every T_2_ progeny transgenic evens of OE8-5 lines were confirmed by *Hpt* PCR (Appendix A).

### 2.3. Overexpression of GmFAD3A Increased Lipid Content

No obvious growth differences were observed between WT and *GmFAD3A*-OE lines at the vegetative growth stage. The *GmFAD3A*-OE lines and WT had the same seed setting rate and grain number per panicle (Appendix A). Because GmFAD3A is an *ω*-3 desaturase, lipids of the seeds were examined. The results indicated that the total lipid contents of *GmFAD3A*-OE lines (OE4-3 and OE8-5) were 11.4% and 40.4% higher than that in WT, respectively (Figure 2c). Combined the high expression levels of two independent transgenic lines (OE4-2 and OE8-5) (Figure 2b), there is co-segregation between *GmFAD3A* overexpression and their phenotypes, indicating that ectopic expression of *GmFAD3A* increased seed lipid contents.

The crude lipid component of rice includes palmitic acid (C16:0), stearic acid (C18:0), oleic acid (C18:1), linolenic acid (C18:3), arachidic acid (C20:0), and several other components. Palmitic acid is the saturated fatty acid with the highest content in rice seeds, and oleic acid is the predominant monounsaturated fatty acid in rice seed oil [16]. Thus, we further tested the main crude FA composition using GC-MS. The results indicated that the contents of palmitic acid, stearic acid, and arachidic acid did not significantly change, whereas the oleic acid decreased (Table 1). There was a substantial increase in the level of oleic acid (+5.6% and +21.5%) and the content of linolenic acid (+25.1% and +47.2%) in *GmFAD3A* transgenic lines compared with WT (Table 1). The total polyunsaturated fatty acids levels in OE4-3 (4.271 mg/g) and OE8-5 (4.983 mg/g) were largely changed compared with WT (3.945 mg/g; Table 1).

### 2.4. GmFAD3A Raised Seed Germination Rate at Low Temperature in Rice.

A previous study showed that stored lipids in seed play a vital role in seed germination and seedling growth [1]. We investigated the seed germination rates of the *GmFAD3A*-OE line (OE8-5) under 15 °C and 28 °C. There was no difference in seed germination rate of *GmFAD3A*-OE lines (OE4-2 and OE8-5) at 28 °C, while seed germination rate of *GmFAD3A*-OE line (OE8-5) (61%) obviously increased under low temperature condition (15 °C) compared to that of WT (about 19.8%) (Figure 3). These results indicated that ectopic expression of *GmFAD3A* enhanced the seed germination rate at low temperature (15 °C) in transgenic rice.

### 2.5. GmFAD3A Enhanced Cold Tolerance in Rice Seedlings

As the seed germination rate of *GmFAD3A*-OE lines was affected by low temperature stress (Figure 3), we designed a set of experiments to validate the roles of GmFAD3A at low temperature. 14-d-old rice seedlings grown under normal conditions were transferred into vermiculite for pot experiments under low temperature treatment (4 °C). Following temperature application and recovery, the WT seedlings withered more seriously, compared with *GmFAD3A*-OE lines (Figure 4). The survival rate of *GmFAD3A*-OE lines significantly increased. After 7 days of low temperature treatment, approximately 23% of WT seedings survived; however, the *GmFAD3A*-OE lines exhibited a 61% survival rate (Figure 4b). In order to analyze the growth of seedlings, the plant height and growth rate of *GmFAD3A*-OE lines and WT were measured. The investigation demonstrated that the plant height of WT showed no visible difference, whereas that of *GmFAD3A*-OE lines increased by 2.03 cm (Figure 4c). Moreover, the growth rate of the *GmFAD3A*-OE seedling was 10.17% which was much higher than that of WT. Taken together, these results confirmed the role of GmFAD3A in cold tolerance.

### 2.6. Determination of MDA, Proline Contents, and Antioxidant Enzymatic Activities

To determine whether ectopic expression of *GmFAD3A* had an effect on the levels of cold stress-related factors (such as MDA and free proline in rice), we tested these factors in WT and *GmFAD3A*-OE seedlings under cold stress and under normal conditions. At 0 h, no obvious differences were found in MDA and proline contents between the transgenic plants and WT. Under cold treatment MDA and proline contents in *GmFAD3A*-OE plants and WT increased. However, MDA content in *GmFAD3A*-OE plants were significantly lower (−33.9% and −43.8%) than that of WT after 1 days and 2 days, respectively (Figure 5a). After 1 h and 2 h treatment, proline contents in WT and *GmFAD3A*-OE plants were raised by 2.2, 17.2 μg/mg and 8.7 and 30.4 μg/mg, respectively (Figure 5b). Compared to in the WT, the proline content in *GmFAD3A*-OE plants was obviously increased. The above results showed that ectopic expression of *GmFAD3A* in rice could promote the accumulation of free proline, leading to a decline in the content of MDA relative to wild type under cold stress.

We also measured the antioxidant enzymatic activities including superoxide dismutase (SOD), peroxidase (POD), and hydroperoxidase (CAT) under low temperature treatment. The result showed that the activities of these three enzymes in leaf tissues increased under cold treatment (Figure 6). Before cold treatment, the activities of SOD, POD, and CAT in GmFAD3A transgenic plants and WT were no difference. However, under cold conditions the SOD, CAT, and POD activities in *GmFAD3A* transgenic lines increased significantly compared with those in WT.

## 3. Discussion

Abiotic stress is a major concern for agriculture worldwide and is responsible for the loss of crop production [17]. Low temperature is an important damaging factor affecting rice growth, development, and productivity in southern China [18,19]. Conventional breeding of cultivated rice has generated increasing yield potential and yield stability. Ectopic expression of the *RsICE1* gene enhanced tolerance to low-temperature stress in rice [20]. In this study, we demonstrated that ectopic expression of GmFAD3A increased lipid contents, and resistance to cold stress in rice. Our results provide a new insight into plant-stress tolerance-related genes and will be useful for improving crop resistance.

In plants, the desaturation of linoleic acid (LA) to ALA occurs in plastids and endoplasmic reticulum (ER). Mutants of linoleate desaturase genes (GmFAD3), decreased the linolenic acid content in commercial soybeans by 98% [10]. Ectopic expression of soybean oleosin genes significantly increased the lipid content in transgenic rice seeds [21]. ER-type *ω*-3 fatty acid desaturase catalyzes the conversion of 18:2 to 18:3 in phospholipids. The lipid transfer resulting from both sets of ω-3 fatty acid desaturases contributes to the total cellular 18:3 content [22]. The ALA contents in the seeds of *GmFAD3A* (an ER-localized gene) overexpressing lines increased significantly [23]. In this study, we also detected a substantial increase in level of oleic acid (+5.6% and +21.5%) and linolenic acid (+25.1% and +47.2%) in *GmFAD3A* transgenic lines compared with WT. However, whether the GmFAD3A protein has a function in rice growth and cold stress in transgenic rice remains a subject for further investigation.

Polyunsaturated fatty acids (PUFAs) are essential for cold acclimation [7]. Genetic manipulation of the levels of PUFAs has led to the modification of cold tolerance in tobacco and poplar trees [7,9]. The unsaturation of lipids has been shown to protect the photosynthetic machinery from photoinhibition at low temperatures [24]. The total polyunsaturated fatty acids levels in *GmFAD3A* transgenic lines (OE4-3: 4.271 mg/g and OE8-5: 4.983 mg/g) were largely changed compared with WT (3.945 mg/g; Table 1). The seed germination rates of *GmFAD3A* transgenic lines increased significantly under low temperature condition (15 °C) compared with WT (Figure 3). Furthermore, the survival and growth rates of *GmFAD3A*-OE seedlings increased significantly compared with WT at 4 °C. This evidence supported the function of GmFAD3A in cold stress.

Cold stress mediates a series of physiological and metabolite changes, such as alterations in chlorophyll fluorescence, electrolyte leakage, reactive oxygen species (ROS), malondialdehyde (MAD), sucrose, lipid peroxides, proline, and other metabolites [19]. The MDA contents in *GmFAD3A* transgenic lines were lower (by −0.03 and −0.04 μmol/mg) than WT under cold stress after 2 d and 4 d, respectively. It was suggested that ectopic expression of GmFAD3A in rice reduced the membrane caused lipid peroxidation caused by cold stress. At the same time, the proline content and antioxidant enzymes activity were observed as being higher than that in the WT under cold stress, which might serve as the frontline of defense against oxidative stress. Thus, we suggested that the ectopic expression of *GmFAD3A* in rice could protect the rice from oxidative damage under cold stress by increasing polyunsaturated fatty acids, antioxidant enzyme activities, and the proline content.

In summary, the data presented here demonstrated that GmFAD3A plays a critical role in cold tolerance and low-temperature germination in rice.

## 4. Materials and Methods

### 4.1. Plant Materials and Growth Conditions

cDNA of soybean (*Glycine max*) was used as template to amplify the full-length coding sequence of *GmFAD3A*. *Oryza sativa* ssp. *japonica* cv. Zhonghua 11 (ZH11) was used as the background. WT (ZH11) and *GmFAD3A* transgenic rice were used for the phenotypic trait analyses. WT and transgenic materials were respectively grown under natural field conditions in Nanchang University, Nanchang, China.

### 4.2. Structure and Sequence Analysis of GmFAD3A

The DNA and protein sequences of GmFAD3A were obtained from Phytozome (https://phytozome.jgi.doe.gov/pz/portal.html) and GenBank (http://www.ncbi.nlm.nih.gov/genbank/), respectively. The genomic sequence and protein sequence of GmFAD3A were analyzed using GSDS (http://gsds.cbi.pku.edu.cn/index.php) and Pfam (http://pfam.sanger.ac.uk/search).

Amino acid sequences of 14 FAD proteins were obtained from rice and soybean. The multiple alignments of amino acid sequences and the neighbor-joining (NJ) phylogenetic tree construction were performed according to previous research [25].

### 4.3. Plasmid Construction and Rice Transformation

The full-length coding region of GmFAD3A was amplified with specific primers FAD3A-C (Appendix A) and cloned into a pCAMBIA1301 binary vector. The construction was sequenced to ensure its integrity. The recombinant vector was maintained in *Escherichia coli* DH5α, and then introduced into *Agrobacterium tumefaciens* EHA105. The construction was transferred into Zhonghua 11 by *Agrobacterium*-mediated transformation, as previously described [26].

### 4.4. RNA Isolation and Quantitative RT-PCR Analyses

The leaves from WT and GmFAD3A transgenic plants used for RNA isolation were frozen in liquid nitrogen and were then stored at −80 °C. Total RNAs were extracted using TRIzol reagent (TransGen, Beijing, China) according to the manufacturer’s protocol. First-strand cDNA was synthesized using PRIME Script Reverse Transcriptase (TaKaRa, Dalian, China, http://www.takara.com.cn/). Quantitative real-time PCR (qRT-PCR) was carried out using an ABI StepOne™ Real-time PCR instrument (Applied Biosystems, Carlsbad, CA, USA, http://www.appliedbiosystems.com/) and Maxima SYBR Green qPCR Master Mix (Thermo, Waltham, MA, USA). Relative expression levels were calculated via the 2^−ΔΔ*C*T^ method [1]. The gene-specific primers used for qRT-PCR are listed in Appendix A.

### 4.5. Analysis of Lipid Content

The crude lipid content was studied using the SoxtecTM 2050 Auto Fat Extraction (Foss^®^ Analytic, Hilleroed, Denmark) according to the Soxtec method [1]. Dehulled rice grains were ground to powder. A quantity of 3.3 g powder was added into the extraction unit, while solvent was added to the extraction cups in a closed system. The extraction consisted of four steps: boiling, rinsing, solvent recovery, and pre-drying. The results were calculated as total amount of fat (g) per 100 g powder. All measurements were performed with three replicates.

The fatty acid (FA) in seeds was extracted using a chloroform-methanol method according to previous study [27]. Quantification of the FA content was performed using GC-MS referring to the method [28]. C13:0 was used as the internal standard. All measurements were performed with three replicates.

### 4.6. Seed Germination Assays

To determine seed germination, 30 seeds of ZH11 and GmFAD3A T_2_ homozygous line (OE8-5) were sterilized and spread on 1/2MS medium. Seeds were placed in a growth chamber with light/dark cycle (12 h/12 h) at 15 °C or 28 °C. Germination was defined as the emergence of the radicles through the seed coat. The germination rate was calculated from the results of three independent experiments.

### 4.7. Cold Stress Tolerance Experiment

For low-temperature treatment, all rice seedlings including ZH11 and GmFAD3A homozygous line (OE8-5) were grown in 1/2 MS medium and normal conditions (28 °C, 16 h light and 8 h dark). 14-day-old seedlings were transferred to a growth chamber at 4 °C for 7 days. The survival ratio, fresh weight, and plant root or shoot length were calculated after stress or recovery in each pot according to the reported methods [29]. 20 independent transgenic plants and 20 ZH11 seedlings were tested in each replicate. Cold treatments were applied in three independent biological replicates.

### 4.8. Physiological and Biochemical Measurements

Samples for physiological and biochemical measurements assay were collected from non-stressed and cold stressed plants. To measure the free proline (Pro) and malondialdehyde (MDA) contents, 0.2 g and 0.5 g of leaves were used for assaying, respectively. All measurements were performed in three biological replicates.

The free proline content was measured by a sulfosalicylic acid method [30]. Quantities of 200 mg (fresh weight) of leaf tissue and 5.0 mL 3% sulfosalicylic acid were added to a mortar for grinding. After centrifugation, 200 µL of supernatant was mixed with 400 µL of glacial acetic acid and 600 µL of 25% ninhydrin and boiled at 100 °C for 40 min. After adding toluene, the absorbance was measured at A520.

The MDA level was studied in reference to the thiobarbituric acid-reactive-substances (TBARS) assay in previous reports [31]. Leaf tissues were frozen in liquid nitrogen and ground to powder. A volume of 1.2 mL of 0.1 (*w*/*v*) trichloroacetic acid (TCA) was added into a tube containing 500 mg of leaf powder, incubated at room temperature for 10 min, then centrifuged at 12,000 rpm for 20 min. An aliquot of the supernatant (0.3 mL) was mixed with 0.3 mL of 0.5% (*w*/*v*) thiobarbituric acid (TBA), and was incubated at 100 °C for 20 min. Then, it was quickly cooled, and was centrifuged at 12,000 rpm for 10 min. The A440, A532, and A600 values of the supernatant were recorded.

### 4.9. Enzyme Activity Assay

For the estimation of antioxidant enzyme activities, 1 g of fresh leaves was ground to powder used liquid nitrogen and then homogenized in 4 mL of chilled buffer. The homogenate was centrifuged for 15 min at 12,000 rpm and the supernatant was collected for various enzymatic assays as the described [32]. The enzymatic activity levels of SOD, CAT, and POD were determined using the method given by Wang [30].

### 4.10. Statistical Analysis

All experimental data were the mean of at least three independent replicates, and comparisons between transgenic and WT plants were performed using one-way ANOVA with Duncan’s multiple range test. All the statistical analyses were performed using SPSS 12.0 software.

## Figures and Tables

**Figure 1 ijms-20-03796-f001:**
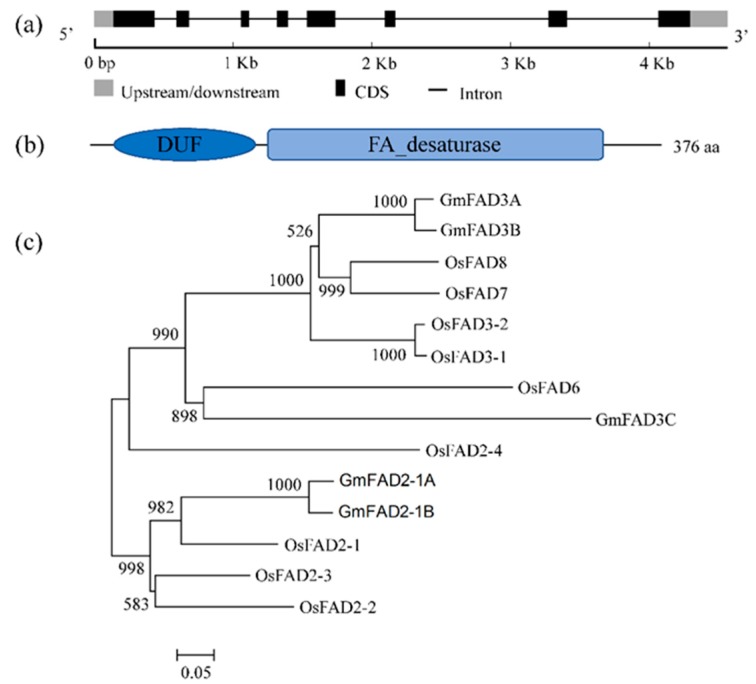
Structural and phylogenetic analyses of GmFAD3A. (**a**) Schematic representation of the exon (black boxes) and intron (intervening lines) organization of *GmFAD3A*. (**b**) A sketch map of protein. The gene encodes a protein of 376 amino acids containing two conserved domains (DUF and FA_desaturase). (**c**) Phylogenic analysis of GmFADs and OsFADs. There were 14 genes chosen from *Oryza sativa* and *Glycine max*.

**Figure 2 ijms-20-03796-f002:**
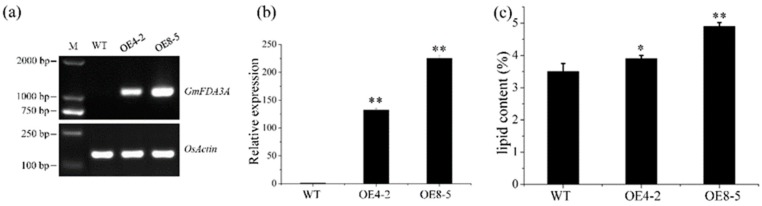
Identification of the GmFAD3A transgenic plants. (**a**) The *GmFAD3A* expression in transgenic plants was determined by semi-quantitation RT-PCR analysis. (**b**) Real-time quantitative PCR analysis in transgenic plants. Values are mean ± SD (*n* = 3). (**c**) Total lipid contents analysis of transgenic rice plants (OE4-2 and OE8-5). Data are mean ± SE for three replicates. *, *p* < 0.05; **, *p* < 0.01.

**Figure 3 ijms-20-03796-f003:**
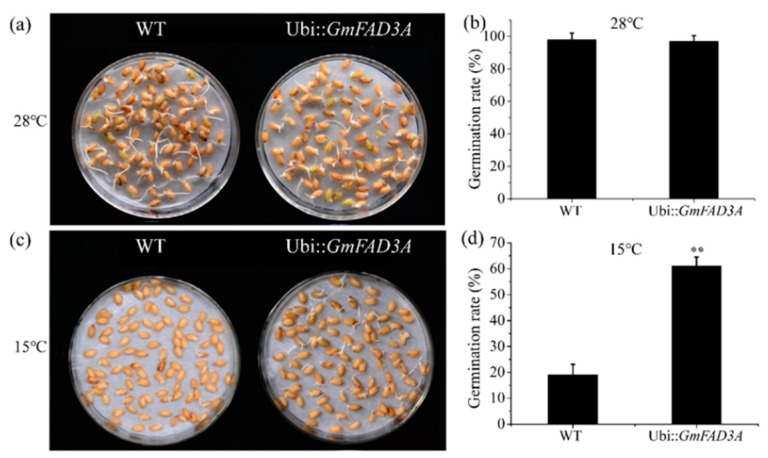
Seed germination assays in mature seeds of wild-type (WT) and *GmFAD3A*-OE line (OE8-5)**.** (**a**,**b**), seed germination of WT. (**c**,**d**), seed germination of *GmFAD3A*-OE line (OE8-5). Germination was defined as the emergence of the radicles through the seed coat. Data are mean ± SE for three replicates with each replicate containing 40 seeds. Data were statistically analyzed vis *t*-test. Asterisks indicate a significant difference between WT and different transgenic lines, ** *p* < 0.01.

**Figure 4 ijms-20-03796-f004:**
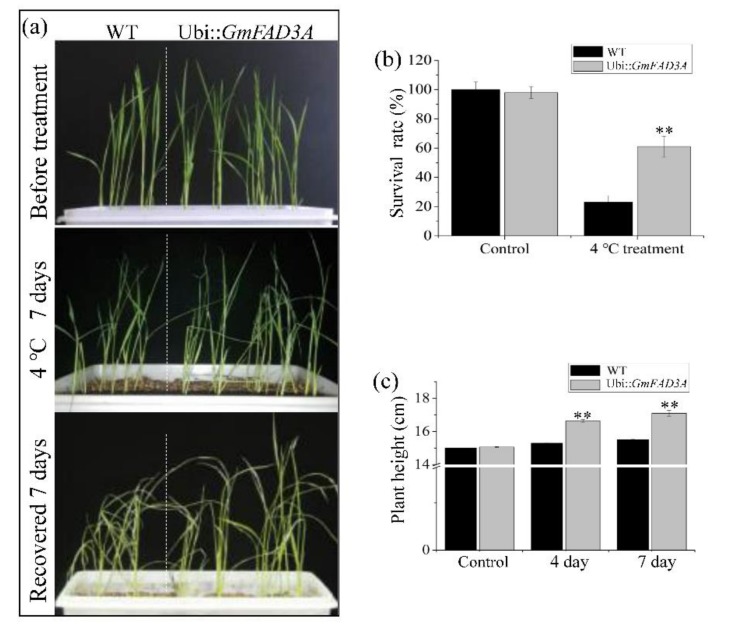
Cold stress tolerance in WT and *GmFAD3A*-OE plants. 14-d-old seedlings were grown under 4 °C for 7 days with a 7-day recovery. (**a**) Representative phenotypes of *GmFAD3A*-OE line and WT. (**b**) Survival rate. (**c**) Plant height. Each biological replicate was the average data collected from 20 plants each of WT and *GmFAD3A*-OE plants. Bars: SD. ** *p* < 0.01.

**Figure 5 ijms-20-03796-f005:**
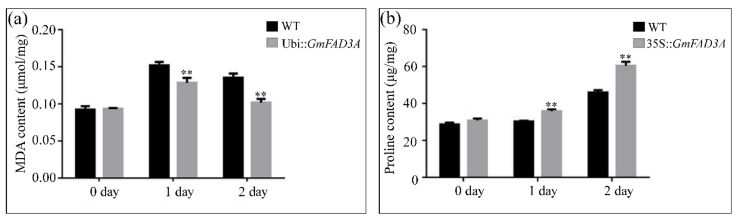
Comparison of MDA and proline content in WT and transgenic line under cold stress. The MDA (**a**) and proline (**b**) contents in *GmFAD3A*-OE and WT plants under cold stress. Data are mean ± SE for three replicates. Bars: SD. ** *p* < 0.01.

**Figure 6 ijms-20-03796-f006:**
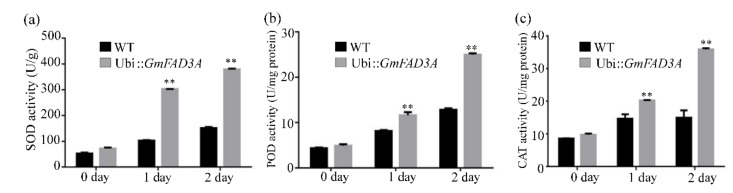
Antioxidant enzyme activity in transgenic and WT plants under cold stress. (**a**) The SOD levels between WT and transgenic plants. (**b**) The POD contents between WT and transgenic plants. (**c**) The CAT activities in WT and transgenic plant. Data are mean ± SE for three replicates. Bars: SD. ** *p* < 0.01.

**Table 1 ijms-20-03796-t001:** Fatty acid compositions in Ubi::*GmFAD3A* T_2_ transgenic lines and WT (ZH11).

Fatty Acid		Content (mg/g)	
	WT	OE4-3	OE8-5
C13:0 (internal standard)	1.982 ± 0.004	1.848 ± 0.005	1.435 ± 0.004
C16:0 (palmitic acid)	1.414 ± 0.056	1.587 ± 0.087	1.566 ± 0.077
C18:0 (stearic acid)	0.263 ± 0.007	0.273 ± 0.009	0.275 ± 0.005
9cC18:1 (oleic acid)	3.468 ± 0.014	3.256 ± 0.018 *	3.186 ± 0.020 *
C18:2 (linoleic acid)	3.842 ± 0.046	4.058 ± 0.016 **	4.669 ± 0.014 **
C18:3n-3 (linolenic acid)	0.127 ± 0.013	0.159 ± 0.015 *	0.187 ± 0.013 **
C20:0 (arachidic acid)	0.058 ± 0.004	0.061 ± 0.005	0.062 ± 0.003
polyunsaturated fatty acids (PUFAs)	3.945 ± 0.026	4.271 ± 0.035 **	4.983 ± 0.054 **

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
