# Peer review of "GmFAD3A, A ω-3 Fatty Acid Desaturase Gene, Enhances Cold Tolerance and Seed Germination Rate under Low Temperature in Rice"

_ijms, 2019, doi:10.3390/ijms20153796_

Round 1

Reviewer 1 Report

The authors Wang et al. investigated the effects of overexpressing GmFAD3a in rice, a fatty acid desaturase from soy bean, driven by a ubiquitin- promotor from maize. They analysed lipid content, stress related enzyme acitvities and germination and growth behaviour in the context of cold stress.

The results in general are very interesting, and several different methods have been applied to determine the effects of gene overexpression on plants treated with cold stress.

The authors worked with several independent transgenic lines, which provides higher reliability of the data.

Statistical analysis has been done well.

My concerns:

The language clearly needs a makeover by a native speaker or a person with a higher level of knowledge of the english language.  

My concerns in detail:

Abstract

Line 15: IS applied

Line 18: it should be from, not form

Introduction:

Line 34: The sentence `Using genetic… lacks some words, please change it. The following sentence `….is an alternative strategy.´ describes the same as the sentence before, I do not see where it presents an alternative strategy. Please clarify this.

Line 41: remove important, they are essential.

Line 46: membraneS, plural.

Line 47: PUFA… The sentence describes the same context as the sentence before, if I am not mistaken.

Line 50: it should be WAS caused….

Line 51: ingredient, singular

Line 53: geneS, plural

Line 54: Mutation IN….

Line 55: Shimada et al., 2000 should have a number.

Line 57/58: sentence However, …. Is not correct, please check.

Line 57: in Reference 13 they used Arabidopsis, not maize, please change it in the text.

Lines 59- 66: presents results, should not be described in the introduction, only in the results or in the summary.

Why did you investigate MDA, SOD and POD? Please write the reasons in the introduction briefly.

Results:

I do not understand if the structural analysis is based on your sequence or on one from NCBI. I do not find a sequence at all under the reference number Glyma.14G194300, and none at all with a length of 4229 bp at all in NCBI. Did you deposit it there or what do you refer to? Please make this more clear. The same is the case for the phylogenetic analysis, it´s not clear to me how you determined exactly these 14 sequences. Did you set a limit of similarity? Or did you search with literature to find the sequences? Please make this more clear in both methods section and figure legends or results. To me this is not clear enough.

Line 87: should be wild type RICE….

Line 89: What is Hpt PCR?

Chapter 2.2:

It is not really clear to me how you proceeded with the transgenic lines. You started with selecting 4 lines, but continued only with 2 of them, is that right? Did the T3 generation include selection for homozygosity? Please clarify. Cause later you talk about homozygous lines, but I cannot see how that was determined.

Figure 2: did you use semiquantitative PCR to determine the expression levels? You are writing about RT- PCR, but that is reverse transcriptase PCR, leading to cDNA. I guess you did a semiquantitative approach, please make this clear to the reader.

In general all Figures 2, 3, 4, 5, 6: you present data in comparison wildtype to transgenic line. I do not understand why now there is always one value for transgenic lines, although two were used. Did you mix the seeds in the end or did you determine a mean value of both lines for all experiments? In general why don´t you present them individually? Are they identical in all experiments? If yes, please indicate it somewhere.

Line 99: it lacks a word behind lipid, maybe `content´?

Line 101: you write about seed setting rate and grain number, but shown in Figure S2 is total spikelets and 1000- grain weight. I assume that is not the same.

Line 102: what do you mean by several major storage substances? If it refers to lipids only, why don´t you write it then?

Lines 104- 106: Where is this correlation between overexpresson and phenoype exactly shown? If it´s a conclusion of all your data, then put it in the discussion, else show a figure here.

Line 106: why do you write `suggesting´? Isn´t that what is shown in figure 2c?

Chapter 2.4: the headline is a whole sentence with the main conclusion of the following paragraph. This is nice, but the other headlines are not like this. Please make this more consistent.

Figure 4b: it should be survival rate.

Why do you present plant height and growth rate individually? If I understand right, then growth rate is the percentage increase in plant height. To me showing plant height is sufficient.

Figure legend 4: you write 20 plants each, in the methods you write 10 plants each.

Material and methods:

Please order the individual chapters in the same order as presented in the results.  

Reviewer 2 Report

The manuscript of Wang et al. describes the effects of over-expression of a soybean w-3 fatty acid desaturase GmFAD3A in rice. Using a battery of molecular and biochemical analyses the authors provide the characteristics of rice phenotypes obtained by GmFAD3Aover-expression, with a special emphasis on cold stress. The provided description includes germination rates, seed lipid content and composition as well as profiling of some secondary metabolites (MDA or ROS) and enzymatic activities in (SOD, POD and CAT) in leaves. The paper reports potentially novel and interesting findings on the role of the fatty acid desaturase in cold stress tolerance in monocot rice. However, meanwhile the experimental part of the paper is of an acceptable quality, the way of description of the result is definitively not. The present version of the manuscript is very chaotic and difficult to interpret for potential readers, especially in terms of unacceptable poor quality of English.

The major points of criticism are:

1.    The substantial language and style correction, ideally by a native speaker, is needed. Below I summarize only some of MANY language and style errors I found in the paper:

p.1, line 15 – “that are applied” should “be that is applied”

p.1, line 16 – polyunsaturated fatty acids (PUFAs) should be used in the whole text

p.1, line 18 – “form” should be changed into “from”

p.1, line 23 – MDA shortcut is used for the first time but it is not explained

p.1, line 23 – “than WT” should be changed into “than in WT”

p.1, line 24 – SOD, CAT and POD are used for the first time in the text but no explanation is given

p.1, line 26 – “suggested” should be changed into “suggest”

p.1, line 35 – What is the point of the sentence: “Using genetic engineering biotechnology to improve cold resistance of rice”?

p.1, line 41 – The sentence: “Lipid and protein are important components of cell membrane” is pointless here.

p.2, line 44 – “In plant” should be changed into “In plants, “

p.2, line 45 – “is essential” (for regulation) is not necessary

p.2, line 46, “for” should be inserted before “transportation”

p.2, line 46, “membrane” should be “membranes”

p.2, line 47, “PUFA” should be “PUFAs”

p.2, line 51, “PUFA is” should be changed into “PUFAs are”

p.2, line 55, “(Shimada et al. 2000)” is not formatted into number

p.2, line 70, “the gene encoded” should be “the gene encodes” 

p.2, line 70, “and” should be inserted before “contained”

p.3, line 75 – what does “a 4, 229-bp coding sequence” means?

p.3, line 87 – all names of bacterial strains should be in italics

p.3, line 89 – what is Hpt standing for?

p.3, line 90, the “ZH11” shortcut is used here for the first time and it is not explained

p.3, line 92, all the comas (,) should be removed from the sentence.

p.3, line 99 – “Overexpression of GmFAD3A increased lipid” – lipid what? content? composition?, saturation?

p.4, line 112, oleic or linoleic?

p.4, line 117, “raised” should be changed into “improves”

p.4, line 118, “played” should be changed into “play”

p.4, line 130, What is the information from the sentence “Statistically significant difference from the wild type within a treatment”?

p.5, line 134, “fourteen-day-old” should be changed into “14 d-old” in the whole text

p.5, line 137, please correct the articles “the” in the whole text

p.5, line 142, there should be a single space between the number and % symbol

p.5, line 144, “Col” should be changed into “Cold”

p.5, line 145, by point a) a word “representative” should be added before phenotype.

p.5, line 148, enzymatic activity of what?

p.5, line 151, the word ”under” should be inserted before “normal”

p.5, line 152, an “plant” should be changed into “plants” and a comma should be placed after the word “treatment”

p6, line 182, please be consistent in the text when using shortcuts (ex. LA, ALA) – explain ALL of them only when used first time and later just use the shortcuts.

p.6, line 183 – it should be “Mutants of linoleate desaturase”

p.7, line 235, “is” should be changed into “was”

p.7, line 241, “its” should be changed into “the”

p.7, line 264 – it should be “was studied in reference to the…”

2.    What was the purpose of analyzing the enzymatic activities of SOD, POD or CAT?

The authors completely miss this information why they did it in the first place.

3.    The same as above refers also to the analysis of the MDA content. What was the point of this experiment? What is the connection between MDA, cold stress and lipid desaturase activity? 

Summing up, this study could potentially be of interest to the scientific community studying plant lipids and abiotic stress response. However, because of many serious shortcomings, and poor presentation of the research idea and style the paper in its present form completely looses its potential. I strongly encourage the authors to resubmit substantially rewritten and corrected version of the paper.